# Does funded research reflect the priorities of people living with type 1 diabetes? A secondary analysis of research questions

Kate Boddy,[1] Katherine Cowan,[2] Andy Gibson,[3] Nicky Britten[1]

[1]NIHR CLAHRC South West Peninsula (PenCLAHRC), University of Exeter Medical School, Exeter, UK
[2]James Lind Alliance, University of Southampton, Southampton, UK
[3]Department of Health and Social Sciences, University of the West of England, Bristol, UK

**Correspondence to**
Kate Boddy;
K.Boddy@exeter.ac.uk

## ABSTRACT

**Objectives** This study explored the divergence and convergence between funded research about type 1 diabetes and the research agenda of people living with the condition and their carers.

**Design, method, setting** A secondary analysis was undertaken of existing data from two UK organisations who regularly work with patients and carers to identify research priorities. The research ideas of people with diabetes were identified in two ways: in 15 research question generation workshops involving approximately 100 patients and carers, and in a James Lind Alliance Type 1 Diabetes Priority Setting Partnership with approximately 580 patients, carers and clinicians (clinician question submissions were excluded from analysis). A total of 859 individual research questions were collected from patients and carers. Diabetes research funding activity was identified through extensive online searches which provided a total of 172 relevant research projects for analysis.

**Results** The data were thematically analysed and areas of priority for research identified and compared between the patient and funded research agendas. The overall finding of this study is that there is substantial convergence between the two research agendas, alongside some important areas of divergence. The key areas of divergence were found in care delivery, injection issues, psychosocial impacts and women's health. We also demonstrate how an apparently convergent priority can host significant differences in emphasis between patient-generated and funded research agendas.

**Conclusions** We offer a comparison of a funded research agenda with one that has been derived directly from people with type 1 diabetes without initial framing by researchers. This provided a rare opportunity to explore the viewpoints of the end-users of research and compare them to realised research as determined by researchers and research organisations.

## Strengths and limitations of this study

► Empirical comparison of the research agendas of people with type 1 diabetes and funded research in this field.
► This is a rare opportunity to explore the research agendas of people with type 1 diabetes, in their own words, through thematic analysis.
► The analysed questions are the suggestions of the people who took part in James Lind Alliance Priority Setting Partnership activities or who took part in a PenCLAHRC (NIHR Collaboration for Leadership in Applied Health Research and Care South West Peninsula) workshop, therefore certain groups could be under-represented and other agendas within the diverse patient communities missed.

## BACKGROUND

The incidence of type 1 diabetes (T1D) is rising globally.[1] As a leading cause of heart disease, stroke and amputation the burden on people living with diabetes and health services is considerable.[2] Well-targeted research has a key role to play in the development of evidence-based treatments for the management and self-management of T1D. People living with diabetes are well placed to contribute to the development of this research in many ways, including suggesting areas for research they consider most important.

A key rationale for patient and public involvement (PPI) is that research carried out with members of the public should lead to research that is more relevant to people's needs.[3] Public involvement in the design and conduct of health research has developed significantly in recent years, and is now seen as a core component of good research practice. Major UK research funders such as the National Institute for Health Research (NIHR) and the Medical Research Council have patient and public representation on funding panels and require funding applications to demonstrate evidence of PPI.[4]

For PPI to be meaningful, patients and the public need to be able to exert influence throughout the research process, including the development of research agendas. Exploring issues of utility and acceptability to

patients before the agenda is set may increase uptake of research,[5] generate research that is of interest to patients, allow outcomes meaningful to patients to be researched and reduce waste.[6 7] INVOLVE, the NIHR's advisory body for public involvement, defines PPI as "research carried out 'with' members of the public rather than 'to', 'about' or 'for' them".[3] Applying this definition of PPI, there are few studies that report involving patients and carers in agenda setting for T1D research. An exception is the James Lind Alliance's (JLA) top 10 research priorities created in partnership with clinicians, patients and carers.[8] Many diabetes agenda setting studies simply omit the patient perspective.[9 10]

Studies that do report patients' diabetes research agendas often do not meet the INVOLVE definition of involvement above since the methods used to draw out these agendas effectively reduce patients to the level of research subjects; the research is 'about' them rather than conducted 'with' them. This is true for surveys, questionnaires and focus groups especially.[11 12] These methods can be employed, as just one stage, within a wider collaborative agenda setting process where involvement exists from beginning to end.[13 14] On their own, however, they offer no opportunity for active involvement because the participants are undertaking agenda setting tasks where there has been prior framing by the researchers. Ultimately, this may contribute to a supposed patient agenda that aligns with that of researchers when in reality a mismatch between the two exists. Further distortion can occur when prioritising research agendas as patient and carer questions may be at greater risk of rejection from the agenda setting process.[15]

It is important then to compare research agendas to ensure that patients' concerns are reflected in ongoing research. Studies comparing the health research agendas of the public with those of researchers are very rare.[16 17] A literature search carried out in relation to this study only located two such comparisons of diabetes research agendas by one author.[12 18] The first study compared patient and researcher preferences and found a mismatch, and the second compared the proportions of research topics and also found that the distribution of funded research did not reflect patient concerns. There is no 'gold standard' currently for producing a 'researcher' derived agenda for comparative purposes. Existing methods include searching for relevant published and unpublished literature,[16] compiling data from ongoing clinical trials[17] and using published abstracts from a relevant conference.[18]

The aim of this study was to explore the divergence and convergence between the agenda demonstrated in T1D research and the research agenda of people with T1D and carers elicited using participatory approaches aligned with the principles of public involvement.[19 20] This provides a rare opportunity for an exploration of an agenda directly expressed by patients and carers in their own words.

## SETTING

This study uses existing data from two UK organisations who regularly work with patients and carers to identify research priorities. The first organisation is the NIHR Collaboration for Leadership in Applied Health Research and Care South West Peninsula (PenCLAHRC).[21] PenCLAHRC involves patients and carers in both the identification and prioritisation of research questions that address patient and clinical concerns.[22] Methods include question generation workshops which enable the public to develop research uncertainties in their own words. Fifteen research question generation workshops were carried out by the PenCLAHRC PPI team between January 2010 and December 2011. These workshops involved approximately 100 patients and carers, including 22 people who identified themselves as either a carer of or person with T1D. Recruitment for the workshops was achieved by sending adverts to local organisations (eg Healthwatch and the Diabetes Association local branch). Workshop attendees were typically retired and white, reflecting the local South West demographic.

The largest proportion of patient questions was provided by the second organisation, the JLA, which is a non-profit making initiative founded in 2004. It brings patients, carers and healthcare professionals together in Priority Setting Partnerships to identify and prioritise the unanswered questions about treatments that they agree are most important.[14] Priority Setting Partnerships work to identify and prioritise the uncertainties, or unanswered questions, about treatments which patients, carers and clinicians want research to address. In the first stage of these partnerships, patient, carer and clinician uncertainties are gathered through a survey where respondents can write their questions in free text, without prior framing by researchers.[23] During the same time period as the PenCLAHRC workshops, research ideas were submitted by people with diabetes to the JLA Type 1 Diabetes Priority Setting Partnership, which was funded by the Insulin Dependent Diabetes Trust. Research ideas were submitted to the Partnership through an online and paper survey, which was widely advertised by various diabetes charities, the Diabetes Research Network, the JLA and NHS Evidence—diabetes. The JLA Type 1 diabetes partnership method for priority setting is described in detail elsewhere.[8]

## METHODS
### Data collection: patients and carers' research questions
Both organisations provided 'raw' data in the form of individual questions, as expressed by the workshop and survey respondents. The data used in this study were received in their original state, prior to any 'cleaning' that the organisations may have done. It is important to note two key points about the data. First, that while the JLA Partnership itself removed questions not related to treatments due to the focus of the JLA method, our study examined all suggestions submitted by people with T1D and carers

in their unedited form, *prior* to any cleaning or rejection (see Snow *et al* for an analysis of the rejected questions[15]). Second, because the analysed data are unedited and directly from the public, some of the questions raised may already be addressed by an existing evidence base. These questions have not been removed, as they were in the JLA process, because they reflect patient concerns.

A total of 877 individual uncertainties were collected from patients and carers. After removing items that were not related to T1D (eg, about type 2 diabetes or an unrelated topic), 859 individual patient questions were available for analysis.

### Data collection: funded research studies

To establish a comparative, researcher-derived data set, we searched for information about research that had been funded within the same time period as the 'patient' agenda setting exercises (January 2010 to December 2011). We defined 'funded research' as UK-based research that was ongoing, for example, a clinical trial in the recruitment phase or research that had been approved for funding by a provider but was not yet under way. By limiting our inclusion criteria to recently funded or newly underway studies we hoped to derive an agenda that was as chronologically close as possible to the patient agenda while still using publicly available existing data. Searches were conducted on 18 websites and databases, identified by an information specialist, to obtain data about diabetes research funded between January 2010 and December 2011 in the UK (see table 1 for details).

The title and abstract of the research records were downloaded providing a total of 231 funded research projects. The projects were screened by KB to ensure that they were about T1D and that the project was solely based in the UK. After duplicates (individual project found repeated through various sources), non-T1D projects (eg, type 2 diabetes) and non-UK projects were removed, 172 unique T1D research projects were available for coding.

Of the 18 sources initially searched, 12 contained information related to T1D research (table 1). To reduce the likelihood of a bias in the research projects analysed, in terms of type of research, extensive attempts were made to cover a full range of research funders including social research funders, rather than simply biomedical research funders (which were easier to locate due to trial registers, and so on). None of the social research funders searched resulted in T1D research projects within the time period of interest although they had previous links with diabetes research which was why they were included in the search process. The breakdown of the number of research projects that were analysed by funder type is shown in table 2.

### Ethics statement

We consider our study to be exempt from requiring ethical approval. The original public involvement agenda setting activities conducted by the JLA and PenCLAHRC, described above, were exempt from requiring ethical

**Table 1**  Sources searched for diabetes research projects funded between January 2010 and January 2012

| Number | Organisation | Codable type 1 diabetes projects |
|---|---|---|
| 1 | The Novo Nordisk UK Research Foundation | 6 |
| 2 | Novo Nordisk | 5 |
| 3 | Diabetes UK | 38 |
| 4 | Diabetes Research & Wellness Foundation UK | 9 |
| 5 | Biotechnology and Biological Sciences Research Council (BBSRC) | 1 |
| 6 | British Heart Foundation | 11 |
| 7 | Chief Scientist Office | 5 |
| 8 | NIHR | 28 |
| 9 | Wellcome Trust | 11 |
| 10 | Medical Research Council | 1 |
| 11 | WHO trials register | 11 |
| 12 | ClinicalTrials.gov | 46 |
| 13 | Picker Institute | 0 |
| 14 | Royal College of Nursing Foundation | 0 |
| 15 | Local diabetes research network contact | 0 |
| 16 | Leverhulme | 0 |
| 17 | Health Foundation | 0 |
| 18 | ESRC | 0 |

approval under guidance published by the NHS National Patient Safety Agency National Research Ethics Service and INVOLVE.[24] The data from those activities are publicly available.[25] This is a secondary analysis of those existing data.

### Data analysis

Both data sets were individually thematically analysed in order to identify inductively emergent research themes.[26] In the early development of this project we opted to employ a form of thematic analysis, over other forms of analysis such as content analysis, because we were interested in understanding the nature of the questions. For each data set, the initial stage of analysis involved familiarisation with the data by KB, resulting in the initial development of a list of recurrent codes. KC also independently undertook an initial analysis of a subset (10%) of both data sets. Specialist knowledge was provided by a consultant endocrinologist who assisted in identifying codes for the funded research data. NVivo, a qualitative software indexing program, was used to facilitate data organisation, coding and retrieval. To develop the codes, each question and research study was considered in terms of its key focus. For example, the question 'Can homeopathic medicines provide beneficial use in diabetic care?' was coded to the

**Table 2** Summary of number of research projects by funder type

| Funder type | Source | Total projects |
|---|---|---|
| Mixed | 1. ClinicalTrials.gov<br>2. WHO—mixed funder register | 57 |
| Diabetes Charity | 1. Diabetes Research & Wellness Foundation UK<br>2. Diabetes UK | 47 |
| UK Government | 1. BBSRC<br>2. Chief Scientist Office<br>3. MRC<br>4. NIHR | 35 |
| Charity (other) | 1. British Heart Foundation<br>2. The Novo Nordisk UK Research Foundation<br>3. Wellcome | 28 |
| Industry | 1. Novo Nordisk (industry) | 5 |

Funder type categories:
*Mixed*: These sources contained multiple types of research funder including charity (diabetes and other), UK government funding and industry.
*Diabetes Charity*: These sources only funded research related to diabetes. Funds are raised through charitable means.
*Charity (other):* These sources funded research into different conditions and types of biomedical research, including T1D. Funds are raised through charitable means.
*UK Government:* These sources funded research into different conditions and types of biomedical research, including T1D. Funds are provided by grants via government research organisations.
*Industry*: These sources funded research into different conditions and types of biomedical research, including T1D. Funds are provided by business and pharmaceutical companies.

category 'Alternative therapies'. KC and KB then met to consult on the codes generated and to discuss, refine and agree to a set of codes for each data set. KB completed the coding of both data sets, in frequent consultation with KC as well as a third researcher (AG). KC and KB met to reach a consensus on the key themes from both data sets. Finally, both data sets were interpreted as a whole, identifying areas of divergence and convergence.

## RESULTS
The results showed that there is convergence between the two research agendas; there are also some notable areas of divergence. We will explore areas of divergence in greater depth than the convergent themes in an attempt to understand what is missing from the agenda from a patient's point of view.

### Areas of convergence
'Control and complications' was an important theme for both the funded research agenda and the patient research agenda. This broad theme relates to all aspects of blood sugar control and complications associated with T1D. Monitoring blood sugar levels was a key area of uncertainty; both agendas were interested in finding better methods of monitoring blood glucose both in terms of providing greater accuracy and in providing more acceptable and less invasive methods. Uncertainties about managing control in particular circumstances, such as exercise, were identified in both data sets. There was also a range of queries about the relationship between maintaining good control and the risk of developing complications:

"Long term effects of maintaining good blood glucose control and its impact on 'diabetic' complications." (People with diabetes research question)

'The objective…is to test whether metformin tablets added in with insulin treatment in type 1 diabetes can prevent the early blood vessel complications which lead to heart attacks and strokes.' (Objective of funded study, WHO)

Another notable area of convergence between the two agendas was themed by the authors as 'causes and cures' for people with diabetes and 'cell research' within funded research. Within the patient research agenda broad questions were posed about finding the causes and cure for T1D: '*How can we cure type 1 diabetes?*' '*What are the causes?*' This may not immediately appear to fit with a funded research title such as '*Development of regulatory B cell assay in type 1 diabetes*' (Title of funded research, Diabetes Research and Wellness Foundation). However, the abstract of this research protocol reveals that there is indeed a shared agenda focused on exploring the causes of diabetes: '*This research will be important for understanding why people develop T1DM.*'

While there are clear differences between the ways in which the funded research questions and people with diabetes express the notion of control and complications, and causes and cures, the data suggested that funded studies are addressing priority areas for the end-users of those studies.

### Areas of divergence
Notable divergent themes were: care delivery, injection issues, psychosocial impacts and women's health.

### Care delivery
The theme of 'care delivery' featured in both agendas, but there was little agreement on the topics to be researched. For people with diabetes there were two aspects of care delivery in need of further research: access to equipment and variation in care. While the first was concerned with the day-to-day mechanisms of managing T1D, both topics were informed by the unequal treatment that people with diabetes can experience, depending on location and local policies. People felt that better provision of essential items, such as blood test strips, would be cost saving in the long term by preventing complications:

'Can we have some conclusive evidence to prove to the nhs that in the long run it is much cheaper to give us more blood test strips than to treat us for the

severe complications that arise when we have poor control through insufficient testing?' (People with diabetes research question)

Variation in care is about the disparity in treatment people with diabetes can face across the country and questions were raised about this unequal delivery of care:

'Why are quality structured education programmes not delivered in all diabetes centres in the UK?' (People with diabetes research question)

The care delivery agenda of the funded research programmes was much broader and no distinct subthemes such as access and variation emerged. There was some overlap with patient concerns. For example, one study aimed to understand the geographical variations in the rates of amputation and foot ulceration in patients with diabetes: '*Social and geographical impact on diabetes related-foot ulcers and amputations*' (Title of funded study, Chief Scientist Office). However, the funded research within the theme of care delivery showed a different emphasis to that of the patient agenda, focusing on areas such as non-adherence and uptake of screening:

'Understanding factors leading to low uptake of diabetic retinopathy screening in Primary Care.' (Title of funded study, NIHR)
'Identifying medication nonadherence: using Scottish routine healthcare data to support improvements in patients care.' (Title of funded study, Chief Scientist Office)

So while 'Care delivery' was prioritised in both funded research and patient questions, there was a clear difference in focus. From the perspective of people with diabetes, lack of service provision is a major concern. From the perspective of researchers, an important concern is services not being used. This demonstrates how an apparently shared priority may host significant differences in emphasis between patient-generated and funded research agendas.

### Injection issues

This theme was clearly important to people with diabetes while notably absent in the funded research. Questions in this theme reflected patient concerns about the need for alternative methods of insulin delivery other than injections. Patients expressed hopes that oral, inhalation or skin patch methods could be developed and prove effective:

'Could insulin be given in tablets instead of having to be injected?' (People with diabetes research question)
'…are there further possibilities that inhalation or any other methods of insulin delivery would work in the future for type 1 diabetes?' (People with diabetes research question)

It is not possible to ascertain from the data whether the question submitters were aware of the existing evidence base and the difficulties in these research fields (eg, limited absorption and enzyme degradation),[27] and are calling for renewed development in these areas or whether they are asking for a novel approach. Be that as it may, our data show that patients want alternatives to be '*quicker and pain free*', '*make life easier*' and be a better method of delivery than the '*very painful and inconvenient*' injections. The extent of these questions and the recurring topic of pain provide insight into outcomes that are important to people with diabetes.

### Psychosocial impacts

The theme of 'psychosocial impacts' of diabetes covered a range of concerns such as peer support, psychological well-being and its effects on diabetes management, and the mental health effects of living with diabetes. This topic was of importance for people with diabetes yet was absent from the funded research agenda. People with diabetes were interested in the way in which psychosocial aspects affect their management and control of blood glucose:

'Does regular contact with others with type 1 diabetes help with blood glucose control and/or mental well-being?' (People with diabetes research question)

They also wanted to know about the way in which living with diabetes affects their psychosocial well-being: '*psychological impact of something that imposes so much on your day to day life*'. Following on from establishing the psychosocial impacts, questions were raised about how to support people: '*What is the most effective support psychologically for people with Type 1 diabetes?*'
In this theme, people with diabetes seemed to be aware of the divergence between their interests and current research. The questions pointed out that this was an area felt by patients to be under-researched and that other topics were given prominence. This was not apparent in other themes:

'The psychological and emotional effect of living with diabetes is chronic <u>and does not seem to be addressed.</u>' (People with diabetes research question)
'<u>From what I have seen research seems to centre around insulin and equipment</u>—I would like to see research around coping mechanisms and how the most successful at dealing with the condition do it.' (People with diabetes research question)

### Women's health

This theme covered a range of gender-specific issues, such as pregnancy, menstruation and menopause. A subtheme, menstruation and menopause, was completely absent from the funded research agenda. Patients were concerned with the effects of menstruation and menopause on diabetes management. Uncertainties centred on how hormone changes brought about by menstruation

and menopause affected diabetes, adding in a layer of additional complications for women:

> 'Hormones have a major effect on diabetes in women. This continues with childbirth and menopause. The full impact of hormones needs to be addressed.' (People with diabetes research question)

> 'What is the effect of the menstrual cycle on Diabetes management?' (People with diabetes research question)

While menstruation and menopause were not addressed by funded research, a second subtheme within 'Women's health', perinatal questions were found in the research set. Research questions addressed in this subtheme covered diabetes management in pregnancy and the effects of maternal diabetes on babies:

> 'The effects of changes in maternal blood glucose concentration on placental blood flow and fetal cardiovascular function during pregnancy in women with Type 1 diabetes.' (Title of funded study, Diabetes UK)

> 'What are the effects of being pregnant on your diabetes and the unborn baby?' (People with diabetes research question)

This divergence around menstruation and menopause, affecting all women with T1D, suggests to us that researchers not only need to talk to patients about their interests and concerns. They also need to ensure they are reaching diverse groups within those patient communities

## LIMITATIONS

This study is based on a relatively large sample of patient and carer questions which have not been modified by researchers, clinicians or policymakers. However, these questions are the suggestions of the people who took part in JLA Priority Setting Partnership activities or who took part in a PenCLAHRC workshop. It might be that certain groups are under-represented and that other agendas within the diverse patient communities have been missed. The study was undertaken at a particular point in time and may not reflect research agendas at other times. While we took steps to ensure the two data sets were as chronologically close as possible, grant applications to funding bodies take time to develop before they ultimately receive funding, meaning that some time lag between the two agendas is inevitable. Finally, none of the research team had live experience of T1D.

## CONCLUSION

Using two robust direct methods of public engagement, this study found an encouraging level of agreement between the research agenda of people with T1D and that of recent funded research. Despite this convergence, important concerns for people with T1D were conspicuously absent from the funded research. Absent topics included questions about the effects of menopause and menstruation on diabetes management and questions about how the pain and impact of injecting could be removed. Also absent were questions about the psychosocial impacts of living with diabetes. Questions within this topic were the only ones that pointed to an awareness of a divergence from the funded research agenda.

The theme of 'care delivery' was present in both research agendas. The funded research was concerned with adherence and screening while the public research agenda focused on the inequalities and variation in care. Our exploration of these research agendas suggests that involvement is needed to ensure that patient concerns are fully reflected within an agenda, avoiding superficial agreement which masks differences of perspective within a topic.

Where people with diabetes have expressed uncertainties that already have an existing evidence base, this suggests that research findings are not reaching the patient community requiring better communication of research and better dissemination to a wider range of audiences. However, if a topic is repeatedly put forward as a priority despite an existing evidence base (eg, oral insulin) consideration needs to be given as to whether those services or treatments are adequately addressing patient needs.

Our findings suggest that funded diabetes research may be neglecting questions about the everyday reality of living with T1D, about quality of life including pain and about the particular needs of women at different stages of the life cycle. They suggest that the diabetes research community could increase the relevance of its work *for* patients by working *with* patients very early in the research development phase.

It is encouraging for the T1D community that our study found considerable agreement between the two agendas. However, the recent paper by Crowe *et al*[17] demonstrates a continued divergence in health research agendas generally. Our work suggests that there may be significant variation between specific conditions and within condition topics. The data generated by these exercises create great potential for marrying up the two agendas and enhancing both communication of convergence and action to address divergence.

The findings of this research contribute to the limited evidence base around public involvement and research agenda setting. We demonstrate the need for considered, meaningful and direct involvement of patients in agenda setting that includes diverse groups within specific patient communities to ensure that the full range of issues experienced by people living with healthcare conditions can be addressed.

**Acknowledgements** We wish to acknowledge the members of the public who took part in PenCLAHRC workshops or the JLA Type 1 Diabetes Priority Setting Partnership.Bijay Vaidya, consultant endocrinologist, for help with understanding aspects of the data. Kristin Liabo, Senior Research Fellow, University of Exeter, for insightful comments on article drafts.

**Contributors** KB and KC have made substantial contributions to the conception and design of the study. They undertook the acquisition, analysis and interpretation of data for the study. They drafted the work and have given final approval of the version to be published. AG and NB have made substantial contributions to the conception and design of the study and interpretation of data. They revised it critically and have given final approval of the version to be published. All authors agree to be accountable for all aspects of the work in ensuring that questions related to the accuracy or integrity of any part of the work are appropriately investigated and resolved.

**Funding** This article presents independent research supported by the National Institute for Health Research Collaborations for Leadership in Applied Health Research and Care in the South West Peninsula (PenCLAHRC). The views and opinions expressed in this paper are those of the authors and not necessarily those of the NHS, the NIHR or the Department of Health. The James Lind Alliance Type 1 Diabetes Priority Setting Partnership was funded by the Insulin Dependent Diabetes Trust charity.

**Competing interests** None declared.

**Provenance and peer review** Not commissioned; externally peer reviewed.

**Data sharing statement** There are no additional unpublished data.

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
