## [Reviewer comments · BMJ Open]

ARTICLE DETAILS

TITLE (PROVISIONAL)	Does funded research reflect the priorities of people living with Type 1 diabetes? A secondary analysis of research questions.
AUTHORS	Boddy, Kate; Cowan, Katherine; Gibson, Andy; Britten, Nicky

VERSION 1 - REVIEW

REVIEWER	Roger Gadsby Honorary Associate Clinical Professor, Warwick Medical School, University of Warwick UK I am lead author of ref 7 which describes the James Lind Alliances research priority setting exercise in Type 1 diabetes
REVIEW RETURNED	20-Mar-2017

GENERAL COMMENTS	This is a very interesting paper in an important field. It is clearly written and well described. There are a few minor alterations which I think would make the paper clearer. 1 Page 3 line 31 "Applying the INVOLVE definition of PPI above" I don't think INVOLVE has been described in the paper and its definition of PPI isn't clearly highlighted in the text. I think the sentence needs expanding to give greater clarity. 2 Similarly p3 lines 38-41 need to be expanded to increase clarity. 3 p4 Lines 22-23 Do we know how many with type 1 attended? If these were general research question generation workshops do we know if people with type 1 diabetes generated the type 1 questions? If most of the questions came from the JLA process what is the added benefit of using the penCLARHRC data?
--

REVIEWER	Jenny Mc Sharry National University of Ireland, Galway, Ireland.
REVIEW RETURNED	27-Mar-2017

GENERAL COMMENTS	OVERALL This study compares the research priorities identified by people with diabetes and their carers to funded diabetes research. This is an important question, in particular with the growing focus on the incorporation of patient agendas in research prioritisation. The manuscript is limited in detail in places, which will reduce the potential use for future researchers and the diabetes community. Below I outline some suggestions on where additional detail would be beneficial.
--

Also, given the limited depth of the qualitative data analysed, could this be supplemented by detail on the numbers under each of the themes in particular when describing the funded research? I wonder if thematic analysis was the best choice of analysis for this study, was content analysis considered?

TITLE

Is qualitative study the best description of the methodology?

ABSTRACT

“alongside some small significant areas of divergence”
What do small significant areas mean here?

BACKGROUND

P. 3 “As a leading cause of heart attacks, stroke and amputation the burden on people living with diabetes and health services is considerable”

Suggest adding reference

P.3 “Applying the INVOLVE definition of PPI above there are few studies that report involving patients and carers in agenda setting for diabetes research”

Suggest providing a brief overview of INVOLVE.

We have done work identifying priorities that included the perspectives of people with diabetes, although our focus was specifically on type 2 diabetes:

Mc Sharry, J., Fredrix, M., Hynes, L., & Byrne, M. (2016). Prioritising target behaviours for research in diabetes: Using the nominal group technique to achieve consensus from key stakeholders. *Research Involvement and Engagement*, 2(1), 14.

P.3 “A literature search carried out in relation to the study we report on here only located two such comparisons of diabetes research agendas by one author [11 15]. The first study compared patient and researcher preferences and found a mismatch and the second compared the proportions of research topics and also found that the distribution of funded research did not reflect patient concerns.”
These two references are particularly relevant to the current study, so it would be useful to provide more detail on these studies including the methodologies used.

P.3 “facilitative methods”

Could you explain what facilitative methods means in this context?

P.4 “The aim of this study was to explore the divergence and convergence between the research agenda demonstrated in funded Type 1 diabetes projects and the research agenda of people with Type 1 diabetes and carers conveyed using facilitative methods.”

Could more of an introduction to the reason for exploring the focus of funded diabetes projects be provided, as this is not mentioned up to this point?

Setting

Could further details (full methods used and participant demographic

info) of the PenCLAHRC workshops be provided as there isn't a reference to a paper with details as provided for the JLA partnership. It would be useful to include demographic details of all participants, as this may impact on type of priorities identified.

METHODS

Was ethical approval obtained from people with diabetes for their priorities to be included as quotes in a published paper?
If yes, included details of ethics, consent procedures etc.

P.5 " Secondly, because the analysed data are unedited and directly from the public, some of the questions raised may already be addressed by an existing evidence base. These questions have not been removed because they reflect patient concerns and show that there is a gap between what is known from research and what is known by patients. Our analysis therefore included topics that were important for patients but not reflected in the funded research because of an existing evidence base"

Not sure that this detail is needed here. Might be enough to state responses are in original state prior to cleaning and to discuss the points above in the Discussion.

P.5 "859 individual patient questions were available for analysis."
Provide breakdown from PenCLAHRC and JLA.

P.5 "Searches were conducted on 18 websites and databases, identified in consultation with an information specialist, containing information about UK research funding related to diabetes (see Table 1 for details)."

Although information is included in Table, suggest adding in timeframe in this sentence.

P.6

"to obtain data about diabetes research funded between Jan 2010 and Dec 2011 in the UK."

What does research funded mean here? Is this projects that were awarded funding, started between these time points or were on-going during this period?

P. 6 "The projects were screened to ensure that they were about Type 1 diabetes and that the project was solely based in the UK."
Who conducted screening, removing of duplicates, coding etc.?

P. 7 Data Analysis

How much data was included in the subset analysis by KC?

How much agreement was there between team members?

What data was included for funded projects? (e.g. title, abstracts, full applications?)

Why was thematic analysis chosen over content analysis for example?

Could a relevant checklist be included (e.g. COnsolidated criteria for REporting Qualitative research)

RESULTS

P.7 Areas of convergence

"The two main convergent themes were 'control and complications' and 'causes and cures'."

What does main mean in this context?

P.8 “the two agendas was expressed as ‘causes and cures’ for people with diabetes and ‘cell research’ within funded research.”
Not clear on meaning of this sentence, expressed by who?

P.9 “This demonstrates how an apparently shared priority may host significant differences in emphasis between patient generated and funded research agendas.”

This is an important and interesting point, suggest including in the abstract.

P.10 “The questions pointed out that this was an area felt by patients to be under-researched and that other topics were given prominence. This was not apparent in other themes”

Again an interesting point, could receive more attention in the Discussion.

LIMITATIONS

“The study was undertaken at a particular point in time, and may not reflect research agendas at other times”

The timing of prioritisation is an important point which should be further addressed as a limitation. The patient generated proprieties were identified in Jan 2010 and Dec 2011 and projects funded between Jan 2010 and Jan 2012 were included.

Grant applications take time to develop, be reviewed and receive funding so even if they were to address patient priorities, they would likely only reflect priorities identified before Jan 2010-Dec 2011. The issues inherent in the time taken in applying for funding and changing patient/healthcare priorities are relevant here and could be discussed.

“Where people with diabetes have expressed uncertainties that already have an existing and adequate evidence base”

Was this a feature of patient identified priorities?

If yes, could examples be provided?

“They suggest how the diabetes research community could increase the relevance of its work for patients [22].”

Unclear of purpose of reference 22 here.

“found high levels of agreement.”

What does this mean in the context of a qualitative analysis?

CONCLUSION

This section is somewhat limited, although this may reflect the suggested limits of the journal.

For example a statement of the principal findings a discussion of strengths and weaknesses in relation to other studies is not included and the meaning of the study in relation to existing literature and the meaning of the study for patients, practitioners, researchers and funders is not discussed in detail.

VERSION 1 – AUTHOR RESPONSE

Reviewer 1		
Comment number	Comment	Author response
1.	This is a very interesting paper in an important field. It is clearly written and well described.	Thank you
2.	Page 3 line 31 "Applying the INVOLVE definition of PPI above" I don't think INVOLVE has been described in the paper and its definition of PPI isn't clearly highlighted in the text. I think the sentence needs expanding to give greater clarity.	Agree. Have described what INVOLVE is and fully quoted definition.
3.	p3 lines 38-41 need to be expanded to increase clarity.	Agree. We have expanded this section to increase clarity and added two references .
4.	p4 Lines 22-23 Do we know how many with type 1 attended? If these were general research question generation workshops do we know if people with type 1 diabetes generated the type 1 questions? If most of the questions came from the JLA process what is the added benefit of using the penCLARHRC data?	22 who identified themselves as either a carer of or person with Type 1 diabetes attended. This number has been added to the text. We consider the added benefit of the PenCLAHRC questions to be a form of validity check. The JLA data and the PenCLAHRC data both allow people with diabetes to pose research questions without prior shaping by researchers but are derived from different approaches – survey and workshops. The PenCLAHRC method produced similar questions to the JLA method in terms of content and subsequent themes. This gives us some assurance, within the limitations set out in the paper, that the questions and derived themes reflect the concerns of people with diabetes in a general sense rather than being a product of the method itself.

Reviewer 2		
Comment number	Comment	Author response
1.	This is an important question, in particular with the growing focus on the incorporation of patient agendas in research prioritisation.	Thank you
2.	The manuscript is limited in detail in places, which will reduce the potential use for future researchers and the diabetes community. Below I outline some suggestions on where additional detail would be beneficial.	Thank you for your detailed review. We hope that by addressing your helpful comments we have strengthened the paper and improved its clarity.
3.	Also, given the limited depth of the qualitative data analysed, could this be supplemented by detail on the numbers under each of the themes in particular when describing the funded research? I wonder if thematic analysis was the best choice of analysis for this study, was content analysis considered?	In the early development of this project we opted to employ a form of thematic analysis, over other forms of analysis such as content analysis, because we were interested in understanding the nature of the questions.
4.	TITLE Is qualitative study the best description of the methodology?	Agree this could be clearer. we have reworded the title as follows: Does funded research reflect the priorities of people living with Type 1 diabetes? A secondary analysis of research questions. We apologise for calling the study qualitative; this was added during the submission process. In essence this study collects existing research questions and uses thematic analysis to facilitate a comparison. It is a secondary analysis and we have removed confusing mentions of the word qualitative.
5.	ABSTRACT “alongside some small significant areas of divergence” What do small significant areas mean here?	Thank you for drawing this to our attention. We agree that this phrasing is misleading and may create unintended meanings for the reader.

		We have changed the wording here and throughout the text to better reflect our meaning.
6.	P. 3 “As a leading cause of heart attacks, stroke and amputation the burden on people living with diabetes and health services is considerable” Suggest adding reference	Have amended text slightly and added the following reference: Centers for Disease Control and Prevention. National diabetes statistics report: Estimates of diabetes and its burden in the United States, 2014. Atlanta, GA: US Department of Health and Human Services; 2014.
7.	P.3 “Applying the INVOLVE definition of PPI above there are few studies that report involving patients and carers in agenda setting for diabetes research” Suggest providing a brief overview of INVOLVE. We have done work identifying priorities that included the perspectives of people with diabetes, although our focus was specifically on type 2 diabetes: Mc Sharry, J., Fredrix, M., Hynes, L., & Byrne, M. (2016). Prioritising target behaviours for research in diabetes: Using the nominal group technique to achieve consensus from key stakeholders. Research Involvement and Engagement, 2(1), 14.	Agree. Have described what INVOLVE is and fully quoted definition.
8.	P.3 “facilitative methods” Could you explain what facilitative methods means in this context?	Agree this is unclear. Have replaced with ‘elicited using participatory approaches aligned with the principles of public involvement.’ The full details of these approaches are detailed in the ‘setting’ section which immediately follows on from this statement. We have also added two references to clarify what principles of public involvement are.
9.	P.4 “The aim of this study was to explore the divergence and convergence between the research agenda demonstrated in funded Type 1 diabetes projects and the research agenda of people with Type 1 diabetes and carers conveyed using	Agree this was not fully explained. We have added a sentence to the background section to introduce the concept of a ‘researcher agenda’ and how this may

	facilitative methods.” Could more of an introduction to the reason for exploring the focus of funded diabetes projects be provided, as this is not mentioned up to this point?	be derived. We have also added detail about what we mean by funded research and the rationale for our choices in the methods section.
10.	Was ethical approval obtained from people with diabetes for their priorities to be included as quotes in a published paper? If yes, included details of ethics, consent procedures etc.	We have added an ethics statement to the methods section explaining why this study is exempt and that it is a secondary analysis of existing data.
11.	P.5 “ Secondly, because the analysed data are unedited and directly from the public, some of the questions raised may already be addressed by an existing evidence base. These questions have not been removed because they reflect patient concerns and show that there is a gap between what is known from research and what is known by patients. Our analysis therefore included topics that were important for patients but not reflected in the funded research because of an existing evidence base” Not sure that this detail is needed here. Might be enough to state responses are in original state prior to cleaning and to discuss the points above in the Discussion.	We prefer to keep this section in the methods as we consider it important information for the reader to have in mind before they encounter the findings. We have reduced its length and removed the more discursive content.
12.	P.5 “859 individual patient questions were available for analysis.” Provide breakdown from PenCLAHRC and JLA.	2% of the questions came from PenCLAHRC.
13.	P.5 “Searches were conducted on 18 websites and databases, identified in consultation with an information specialist, containing information about UK research funding related to diabetes (see Table 1 for details).” Although information is included in Table, suggest adding in timeframe in this sentence.	Have merged two sentences so that the time frame information is now more clearly available.

14.	P.6 “to obtain data about diabetes research funded between Jan 2010 and Dec 2011 in the UK.” What does research funded mean here? Is this projects that were awarded funding, started between these time points or were on-going during this period?	See above response – have added definition and expanded.
15.	P. 6 “The projects were screened to ensure that they were about Type 1 diabetes and that the project was solely based in the UK.” Who conducted screening, removing of duplicates, coding etc.?	KB conducted the screening and is a qualified information specialist. Detail added to text.
16.	How much data was included in the subset analysis by KC?	We have added the figure to the analysis section.
17.	How much agreement was there between team members?	Differences in initial coding undertaken by KB and KC were discussed in order to clarify meaning and ensure consistency.
18.	What data was included for funded projects? (e.g. title, abstracts, full applications?)	Have added details to the methods section to clarify.
19.	Why was thematic analysis chosen over content analysis for example?	Please see response above to comment 3.
20.	Could a relevant checklist be included (e.g. COnsolidated criteria for REporting Qualitative research)	We have amended the text in several places to clarify that this is a secondary analysis and removed references to it being a qualitative study.
21.	P.7 Areas of convergence “The two main convergent themes were ‘control and complications’ and ‘causes and cures’.” What does main mean in this context?	Agree this is misleading. We have changed this phrasing here and throughout, as per comment 4 above.
22.	P.8 “the two agendas was expressed as ‘causes and cures’ for people with diabetes and ‘cell research’ within funded research.” Not clear on meaning of this sentence, expressed by who?	Have removed ‘expressed’ and clarified by adding “were themed by the authors”.
23.	P.9 “This demonstrates how an apparently shared priority may host significant differences in emphasis between patient generated and funded research agendas.” This is an important and interesting	Thank you, we have added this to the abstract.

	point, suggest including in the abstract.	
24.	P.10 “The questions pointed out that this was an area felt by patients to be under-researched and that other topics were given prominence. This was not apparent in other themes” Again an interesting point, could receive more attention in the Discussion.	We are constrained by the word limit. However we have added a sentence to the conclusion to emphasise this finding.
25.	“The study was undertaken at a particular point in time, and may not reflect research agendas at other times” The timing of prioritisation is an important point which should be further addressed as a limitation. The patient generated proprieties were identified in Jan 2010 and Dec 2011 and projects funded between Jan 2010 and Jan 2012 were included. Grant applications take time to develop, be reviewed and receive funding so even if they were to address patient priorities, they would likely only reflect priorities identified before Jan 2010-Dec 2011. The issues inherent in the time taken in applying for funding and changing patient/healthcare priorities are relevant here and could be discussed.	We have added a sentence to the limitations to expand this point.
26.	“Where people with diabetes have expressed uncertainties that already have an existing and adequate evidence base” Was this a feature of patient identified priorities? If yes, could examples be provided?	We have provided an example in the injection issues theme discussion and a supporting reference. We have clarified and expanded further in the conclusion.
27.	“They suggest how the diabetes research community could increase the relevance of its work for patients [22].” Unclear of purpose of reference 22 here.	Thank you for picking up this error – reference added in error.
28.	“found high levels of agreement.” What does this mean in the context of a qualitative analysis?	Agree misleading – have amended.
29.	This section is somewhat limited, although this may reflect the suggested limits of the journal.	We have expanded two sections within the conclusion whilst being mindful of the word

	For example a statement of the principal findings a discussion of strengths and weaknesses in relation to other studies is not included and the meaning of the study in relation to existing literature and the meaning of the study for patients, practitioners, researchers and funders is not discussed in detail.	limit.
--	---	--------

VERSION 2 – REVIEW

REVIEWER	Prof Roger Gadsby Honorary Associate Clinical Professor, Warwick Medical School, University of Warwick
REVIEW RETURNED	29-May-2017

GENERAL COMMENTS	The authors have responded appropriately to the questions from the reviewers and as a result have incorporated changes to the paper which make it more clear and understandable.
--

REVIEWER	Jenny Mc Sharry National University of Ireland, Galway Ireland
REVIEW RETURNED	25-May-2017

GENERAL COMMENTS	The changes made have made the paper clearer and will increase its usefulness to researchers, funders and the public. The change to the title in particular is a more accurate reflection of the content. One final suggestion, could the justification for the use of thematic analysis provided in the response to reviewers be incorporated into the manuscript?
---